# Quantitative Methods for the Prioritization of Foods Implicated in the Transmission of Hepatititis E to Humans in Italy

**DOI:** 10.3390/foods11010087

**Published:** 2021-12-29

**Authors:** Ornella Moro, Elisabetta Suffredini, Marco Isopi, Maria Elena Tosti, Pietro Schembri, Gaia Scavia

**Affiliations:** 1Department of Food Safety, Nutrition and Veterinary Public Health, National Institute of Health, 00161 Rome, Italy; elisabetta.suffredini@iss.it (E.S.); gaia.scavia@iss.it (G.S.); 2Department of Mathematics, University of Rome, Piazzale Aldo Moro 5, 00185 Rome, Italy; isopi@mat.uniroma1.it; 3National Center for Global Health, National Institute of Health, 00161 Rome, Italy; mariaelena.tosti@iss.it; 4Regional Department for Health Activities and Epidemiological Observatory of the Sicilian Region, 90145 Palermo, Italy; p.schembri@regione.sicilia.it

**Keywords:** hepatitis E virus, epidemiology, mathematical modeling, food safety

## Abstract

Hepatitis E is considered an emerging foodborne disease in Europe. Several types of foods are implicated in the transmission of the hepatitis E virus (HEV) to humans, in particular, pork and wild boar products. We developed a parametric stochastic model to estimate the risk of foodborne exposure to HEV in the Italian population and to rank the relevance of pork products with and without liver (PL and PNL, respectively), leafy vegetables, shellfish and raw milk in HEV transmission. Original data on HEV prevalence in different foods were obtained from a recent sampling study conducted in Italy at the retail level. Other data were obtained by publicly available sources and published literature. The model output indicated that the consumption of PNL was associated with the highest number of HEV infections in the population. However, the sensitivity analysis showed that slight variations in the consumption of PL led to an increase in the number of HEV infections much higher than PNL, suggesting that PL at an individual level are the top risky food. Uncertainty analysis underlined that further characterization of the pork products preparation and better assessment of consumption data at a regional level is critical information for fine-tuning the most risky implicated food items in Italy.

## 1. Introduction

Hepatitis E virus (HEV) is the causative agent of hepatitis E, an emerging disease of worldwide occurrence affecting humans. The clinical course of hepatitis E is frequently asymptomatic. Clinical signs include fever, anorexia and jaundice. Extrahepatic manifestations and serious sequelae, including chronic condition leading to liver failure and death, may occur especially in immunocompromised patients and in the presence of comorbidity [1,2,3,4,5]. A major risk for chronic and fulminant hepatitis E is reported for pregnant women, with the possibility of abortion or infant mortality [6].

For many decades, in Europe and in the USA, hepatitis E was considered a health problem limited to travelers coming back from areas where hepatitis E was endemic [7,8]. However, since the early 1990s, autochthonous cases have been increasingly reported [9]. In Europe and other high-income countries, hepatitis E is considered a foodborne zoonosis causing mainly sporadic cases [7,8,10]. Outbreaks of HEV infection were also occasionally reported in the European Union (EU) [11,12,13]. In the EU, however, surveillance of HEV infection is sparse and not harmonized, hampering the possibility to adequately characterize the epidemiology of hepatitis E, including the accurate identification of the food items implicated in the transmission of HEV to humans.

HEV genotypes affecting humans mainly belong to HEV3 and HEV4. These genotypes are frequently detected in pigs and wild boars, which are considered the main HEV animal reservoirs [3]. The majority of human cases are attributable to consumption of pork and wild boar meat and products thereof [2,14,15]. Likewise, in Italy, evidence from several studies indicates that the consumption of raw or undercooked pork and wild boar meat is an important risk factor [14,15]. Some studies also reported shellfish and leafy vegetables as vehicles potentially implicated in foodborne transmission of HEV [16,17,18,19]. Evidence is also available of a higher risk for professionally-exposed workers such as veterinarians, farmers and hunters [20,21,22,23].

Circulation of HEV in farmed pigs in Italy is widely documented [24,25,26,27,28]. In the food chain, HEV has been detected in pork foods such as dry and fresh sausages at retail level [29,30,31] but also in shellfish sampled in the production areas or in biomonitoring points [32,33], at retail [34] or in biomonitoring points [35]. Observational studies conducted in several countries also documented HEV contamination of vegetables and fruits [18,36,37,38]. HEV RNA was found in sewage and surface water samples, suggesting possible environmental contamination via recycled water [33,39,40].

Understanding the dynamics and transmission of HEV from animal reservoirs to humans and of food contamination is key to reducing the incidence of hepatitis E in the population through the adoption of specific control actions in the food production chain. In recent years, several studies modeled the dynamic of HEV spread along the food chain in different EU countries at both the pre-harvest and post-harvest level, including consumer exposures [41,42,43,44,45].

In our study, we developed a mathematical model to rank the importance of various types of food potentially implicated in the transmission of HEV to humans in the adult Italian population (around 50,000,000). The food categories considered in our study are pork products with liver (PL), pork products without liver (PNL), bivalve shellfish (SH), green leafy vegetables (GV) and raw milk (RM).

## 2. Methods

In order to obtain the ranking of food items most frequently implicated in HEV transmission in Italy, we developed a parametric stochastic model to estimate the expected number of newly infected persons who develop HEV infection in the Italian population (≥18 years) through the consumption of the different foods in a one-year period.

The analyses were carried out with the R software version 3.6.0 [46]. For the heaviest calculation, the Gauss Cluster at the Turing Lab of Mathematics Department “Guido Castelnuovo” of Rome “La Sapienza” was used (http://centrocalcolo.mat.uniroma1.it, http://turinglab.mat.uniroma1.it, accessed on 1 November 2021).

### 2.1. The Mathematical Model

We modeled the individual infectious dose distribution *S* to HEV using a proxy of the infectious dose based on data available in the literature. We thus built the distribution $C_{i}$ of HEV concentration in a food serving of category *i*, based on data on prevalence of HEV contamination of food at retail obtained from a recent sampling study in Italy. Using these two quantities, we estimated the probability $q_{i}$ for a single person to develop a HEV infection after the consumption of a single serving of food belonging to category *i*. The average number of portions of each food consumed per year per person and over the number of susceptible individuals in the Italian population (Figure 1) were then summed up to estimate the average number of newly infected cases in the Italian population in one year. All the parameters used in the model are summarized in Table 1.

The model framework is showed in Figure 1.

We built the distribution of concentration per serving $C_{i}$ for each food category using HEV load data in food expressed in genome equivalent per gram of food (g.e./gram) (see Section 2.2), and defining a mean serving size $serv_{i}$ (grams) for each food category, according to guidelines of Council for Agricultural Research and Analysis of the Agricultural Economy [51]. Multiplying these two quantities, we obtained the distribution of the total genome equivalent HEV per serving (g.e./serving). The distribution was built as a mixture of random variables as follow
$$C_{i}=\alpha_{i}C_{0}^{i}+\beta_{i}C_{1}^{i}\;\mathrm{where}\;C_{0}^{i}∼\delta_{0},\;C_{1}^{i}∼exp\left(\lambda_{i}\right)$$
$C_{0}^{i}$ is a Dirac delta with point mass in zero and represents the HEV negative food samples belonging to category *i*. The random variable $C_{1}^{i}$ models the distribution of the viral concentration in the HEV positive samples belonging to category *i* and it is distributed according to an exponential distribution. The rate of the exponential distribution is the inverse of the mean viral concentration $\lambda_{i}^{-1}$ of HEV positive samples that we estimated using the maximum likelihood estimation (MLE). The weights $\alpha_{i}$ and $\beta_{i}$ are the fraction of HEV negative and positive food samples, respectively.

To model the individual infectious dose distribution *S*, we used data from outbreaks for which the HEV load in implicated food (g.e./gr) was documented [13,47,48,49]. We fitted *S* on the estimated foodborne HEV intake (g.e.) of cases involved in these outbreaks. The individual intake of HEV (g.e.) was estimated based on the viral concentration in the implicated food (HEV g.e./gr) for a mean serving size (gr) of the implicated food. The mean serving size ($serv_{0}$) was estimated according to the same data source used for $serv_{i}$. We modeled *S* as an exponential distribution with parameter μ, which was estimated using MLE.

Any exposure to HEV, through the consumption of a single food serving, leading to an intake of HEV g.e. higher than the individual infectious threshold was defined as a new HEV infection event. We defined $q_{i}$ as the probability of infection given the consumption of one single food serving. Each foodborne exposure to HEV was considered independent and no cumulative exposure to HEV in multiple meals was assumed possible.

Based on the available data on food consumption in Italy, we estimated the number of average servings consumed in one year per person $d_{a}^{i}$, for each food category *i*. Considering each meal as a Bernoulli trial of probability $q_{i}$, we built a geometrical random variable *T* of parameter $q_{i}$ modeling the number of failures before the first successful exposure (i.e., infection). We estimated the probability $p_{i}$ for an individual to become infected in one year as
$$p_{i}=P\left(T\leq d_{a}^{i}\right)$$
meaning the probability that the first successful exposure happens within one single year. The number *N* of susceptible individuals in the Italian adult population was estimated subtracting to the total Italian population the fraction of HEV seropositive individuals. This latter fraction was estimated based on data from a HEV seroprevalence survey among blood donors in Italy. We assumed a long-life immunization status against HEV of seropositive subjects (i.e., no reinfection possible). Moreover, we assumed immune individuals homogeneously distributed all over the Italian population. We considered the total number of new infections per year *X*, given the consumption of food belonging to category *i*, to be binomially distributed with parameters $\left(N,p_{i}\right)$. Whence, we obtained the average number of new infected individuals in one year, as the expected value of *X*, meaning $N_{inf}^{i}=E\left(X\right)=Np_{i}$.

### 2.2. Data and Data Sources

Data on HEV prevalence and concentration in food were generated from a sampling study carried out in Italy between 2016 and 2019 within the project “CCM 2016—Hepatitis E, an emerging problem in food safety” (Suffredini, personal communication). Briefly, 730 samples were collected at retail level in different areas of Italy: North, Center and South (see Table 2). Samples were analyzed using matrix-specific viral concentration procedure for pork products [52], bivalve shellfish, green leafy vegetables (ISO, 2017) and raw milk [12]. Detection and quantification of HEV in samples was performed by real-time RT-qPCR as detailed in Di Pasquale et al. [53].

The number of individuals susceptible to HEV in the Italian population was estimated based on official demography data (as of 1 January 2021) (ISTAT (http://dati.istat.it/, accessed on 6 December 2021) and on a HEV seroprevalence study conducted among 10,011 blood donors’ plasma unit samples (≥18 years) in 2018 (0.02% of the adult Italian population as of 31 December 2018) [50].

The number of food servings consumed in one year by a single person was estimated based on food consumption data sourced from a nation-wide consumption survey conducted in Italy in 2005–2006 [54], available on FAO/WHO GIFT tool platform (http://www.fao.org/gift-individual-food-consumption/en/, accessed on 1 November 2021). Details are reported in Appendix A.

Data collected during outbreak investigations with implicated food analysis were used to estimate the dose–response curve [13,47,48,49].

#### 2.2.1. Uncertainty and Sensitivity

We performed uncertainty and sensitivity analysis to quantify the variability in the output due to the variability in the input parameters [55]. We followed a sampling-based method as described in Saltelli [56,57] and generated 10,000 samples for each estimated parameter (i.e., $cons_{day}^{i}$, λ and μ) using parametric bootstrap [58,59] and ran the model obtaining a sample of the same size for the output. This output sample was used to quantify the uncertainty and to perform the sensitivity steps with the aim to explore the effect of each parameter on the model output. We followed a two-step approach. We first analyzed the scatterplots of input parameters versus the output and then calculated the standardized regression coefficient for each of the parameters by linear regression analysis. This latter step gave us the metric to rank the parameters. Details are described in Appendix C and Appendix D.

#### 2.2.2. Evidence from the Italian Surveillance System

Information on food consumption in cases of hepatitis E reported to the Italian surveillance system for acute viral hepatitis (SEIEVA) between 2016 and 2019 was used to discuss the outputs of the model. The SEIEVA is a voluntary system set up in 1985 by the Italian National Institute of Health, now covering 83% of the Italian population [60,61]. Since 2007, the local health units voluntary participating in the surveillance are required to perform and report HEV laboratory testing. Hepatitis E case definition is based on the positivity to IgM anti-HEV antibodies and elevate serum transaminases level (with or without clinical symptoms). Since the start of the SEIEVA activities, information on risk factors including food exposures were collected using a standardized questionnaire for all cases of acute viral hepatitis. The questionnaire was revised and released to include specific hepatitis E risk factors in late 2016. This activity was also completed within the national project “CCM 2016—Hepatitis E, an emerging problem in food safety”.

## 3. Results

### 3.1. Parameter Estimation

Food-specific parameter estimations are reported in Table 3. We also reported a 95% interval confidence for the parameters estimated directly from data.

The results of the food sampling survey indicated that the highest proportion of HEV positive samples (i.e., HEV food prevalence $\alpha_{i}$) belongs to PL products (11%), followed by PNL (2.8%) and SH (0.48%). No positive samples were found for the GV and RM categories (0% prevalence) (Table 3).

The number of expected genome equivalents per 100 gr size of serving (i.e., $\lambda_{i}^{-1}=E\left(C_{1}^{i}\right)$) for PL and PNL products were $7.8\times 10^{4}$ g.e./serving and $3.4\times 10^{4}$ g.e./serving, respectively. The average servings consumed per person per year was 3 and 73, respectively, for PL and PNL. The only positive sample for SH resulted in $8.7\times 10^{4}$ g.e. per a 150 g average serving size. The expected number of SH servings consumed yearly are 26. For GV and RM categories, we did not obtain results for $\lambda_{i}$ because no positive samples were detected. These estimates are shown in Table 4, where all the parameters directly involved in the ranking activity are also reported. We included in Appendix E a risk matrix that uses these parameters to also profile a qualitative ranking of the food categories.

The other parameter estimations are reported in Table 5. The average serving size consumed at outbreak events $serv_{out}$ resulted to be 100 gr, yielding to a mean individual infectious dose of $7.3\times 10^{6}$ g.e. (i.e., $\mu^{-1}$). The proportion of seropositive individuals of Italian population *h* was derived by Spada et al. [50] and resulted to be 8.3%. We subtract this proportion to the total Italian population to obtain the amount of susceptible individuals. The total population as of 1 January 2021 was estimated to be 50,208,329, yielding to 45,840,205 susceptible individuals. All the detail on methodological aspects of parameters estimation are reported in Appendix B.

### 3.2. Model Output

For each food category, the outputs of each step of the model are presented in Table 6, including individual probability of infection following the consumption of a single serving $q_{i}$, individual probability of infection in a year $p_{i}$ and the expected number of new infected per year $N_{inf}^{i}=E\left(X_{i}\right)$.

In addition, we reported density and cumulative probability sketches for new HEV infected individuals per year $X_{PL}$, $X_{PNL}$ and $X_{SH}$ (Figure 2). Standard deviations of these three binomial random variables are 409 for PL, 668 for PNL and 260 for SH.

### 3.3. Uncertainty Analysis

We obtained bootstrap samples from input parameters and model output as described in Section 2.2.1. We considered parameters uncorrelated given the Pearson correlation test results that are reported in Appendix C, where standard deviations of input parameter samples are also shown (see Table A1).

In Table 7, we reported summary statistics of the output distribution for the three categories involved in the analysis. In Figure 3, Figure 4 and Figure 5, we displayed the histogram and the sample cumulative distribution of the output samples for category PL, PNL and SH, respectively. The output shown for this analysis is the individual infection probability in a year $p_{i}$.

### 3.4. Sensitivity Analysis

From uncertainty analysis, we obtained samples of input parameters and output used to perform the sensitivity analysis. The three scatterplots (Figure 6, Figure 7 and Figure 8) show the variations of the output samples versus the three parameters involved in the analysis, underlying the strength of the dependencies between them. The shape $cons_{day}^{i}$ plot is a bit flattened on the bottom with no strong structure for all the food categories, suggesting a very light dependency between the output and this parameter. λ seems to have a little more structured outline, especially for the PL category, but the output exhibits the strongest dependency with the μ parameter whose plots shows a well-defined shape, especially for PL and PNL.

To quantify the relative importance of input parameters, we conducted a regression analysis using $\lambda_{i},cons_{day}^{i}\;\mathrm{and}\;\mu$ as covariates. This, as explained in [57] (Cap.1), allowed us to rank these three input parameters based on their impact on the output. Results are reported in Table 8, Table 9 and Table 10.

As already suggested by scatterplots, μ was the parameter that most influences the output for both the categories PL, PNL. The overall influence order for each category is the following:$$|\hat{\mu }\left|>\right|\hat{cons_{day}^{PL}}\left|>\right|\hat{\lambda_{PL}}|$$
and
$$|\hat{\mu }\left|>\right|\hat{\lambda_{PNL}}\left|>\right|\hat{cons_{day}^{PNL}}|$$

From the value of the standardized estimates of the coefficients, we can evaluate the impact of each of them for perturbations equal to a fixed fraction of parameter’s standard deviation [56] (Cap. 6). For the PL category, μ has an impact 4% higher than $cons_{day}^{PL}$ and 14% higher than $\lambda_{PL}$, while $\lambda_{PL}$ has an impact 11% higher than $cons_{day}^{PL}$. For the PNL category, μ has an impact 55% higher than $\lambda_{PNL}$ and 68% higher than $cons_{day}^{PNL}$. The impact of $\lambda_{PNL}$ is about 29% higher than $cons_{day}^{PNL}$. For the PL category, it is clear that $cons_{day}^{PL}$ and $\lambda_{PL}$ are very close and all the three parameters have a relatively small impact.

The $R^{2}$ statistics for both categories are high, 0.97 for PL and 0.83 for PNL, meaning that these three parameters account for almost the entire uncertainty in the output.

A slightly different pattern is shown by SH category, where
$$|\hat{\mu }\left|>\right|\hat{cons_{day}^{SH}}\left|>\right|\hat{\lambda_{SH}}|.$$

We found that μ has an impact 98% higher than λ and 92% higher than $cons_{day}^{SH}$, for a perturbation equal to a fixed fraction of parameter’s standard deviation. The $R^{2}$ for this parameter resulted much smaller with a 0.51, suggesting that other factors are contributing to the uncertainty. This is not surprising given that we have only one positive sample for this category.

Last, in order to have a final quantification of the stability of the output, we made a simple experiment on $d_{i}^{a}$ parameter (so, indirectly on $cons_{day}^{i}$). We increased the $d_{PL}^{a}$ to actually see the results in number of exposed people per year. We observed that with $d_{PL}^{a}=6$ the number of infected people reached 335,446 and with $d_{PL}^{a}=8$$N_{inf}^{PL}=446,715$.

More details on results of these analysis are reported in Appendix C.

### 3.5. Food Consumption in Italian Hepatitis E Cases

Between 2016 and 2019, a total of 213 autochthonous cases of hepatitis E were reported to the SEIEVA system. The availability of information on foods consumed by patients before the onset of the disease varied considerably among the different foods, depending on the type of questionnaire administered to hepatitis E patients. Information on shellfish consumption was available for a high proportion of cases (N = 186; 87%) because the exposure to this foodstuff was investigated through both the general questionnaire for acute viral hepatitis and the specific questionnaire for hepatitis E, in place from late 2016. For all the other food items, which were investigated with the hepatitis E questionnaire only, this proportion did not exceed the 43% of the hepatitis E cases (Table 11), making the uncertainty around the exposure prevalence to these foods much higher. Pork meat and pork cured meat were by far the food items most frequently consumed by hepatitis E cases, with more than 69% of the patients having consumed these foods, followed by pork liver, fruits, shellfish, vegetables and wild boar meat (Table 11).

## 4. Discussion

The model output shows that the consumption of PNL led to the greatest exposure to HEV in the Italian population and was associated with the highest number of new expected HEV infections per year, followed by the consumption of PL and SH. Based on these findings, the risk posed by PNL is ranked first at the population level among foods implicated in the transmission of HEV, followed by PL and SH. For the other foods considered by our study (i.e., GV and RM), no expected cases of HEV infections were estimated by our model. The consumption of pork products has been frequently indicated as a risk factor for foodborne HEV infection [62]. This type of food has also been frequently linked to foodborne outbreaks of hepatitis E [47,48,63,64]. The consumption of shellfish has also been pointed out as a possible risk factor in some studies [48,65], although, to our knowledge, no outbreaks implicating the consumption of contaminated shellfish have ever been reported in Europe.

PNL are consumed much more frequently than PL and SH and by a larger proportion of population. This explains why the highest expected number of new cases in the population is associated with this food despite the mean prevalence of HEV contamination and the viral load per serving being higher for PL and SH.

The sensitivity analysis indicates that even a slight increase in the consumption of PL servings at the individual level results in a remarkable increase in the expected number of new cases of HEV infection. As an example, passing from three to eight portions of PL consumed per person per year, which is a realistic variation at individual level, the number of new HEV infections in the population increases from about 168,000 to approximately 450,000 cases, revealing that the number of servings of PL at the individual level is a critical element to be taken into account for food risk ranking. Similar variations in the consumption of PNL do not result in comparable increases in the number of new HEV infection cases in the population. These findings provide evidence of the importance of collecting very accurate data on PL consumption at the population level in order to strengthen the HEV food risk ranking and highlight the importance of PL consumption for the risk of HEV transmission at the individual level.

The consumption of PL in Italy varies hugely among both individuals and population subgroups, depending not only on personal preferences but also on traditional differences in consumption habits. There is a wide geographic variability in the recipes and mode of preparation of pork products, with important local peculiarities, especially for products such as cured meat and offal. This may lead to highly heterogeneous consumption of PL and exposure to HEV in specific population subgroups and geographical area. Unfortunately, the food consumption data source used in our study lacks sufficient details to allow for more accurate estimations of HEV exposure associated with the different types of PL.

Food consumption data from hepatitis E cases reported to the SEIEVA (see Section 3.5) are too scarce to support a formal validation process of our model. However, they are in line with the evidence obtained in our study about the importance of both PNL and PL as top risk foods for the transmission of HEV to humans. The consumption of pork meat and pork cured meat was reported by a high proportion of cases (>70%), while pork liver as well as shellfish by a much lower proportion of cases (i.e., each food group not exceeding 30% of the cases). Although the SEIEVA data would indicate that a minor proportion of hepatitis E cases had consumed pork liver, it is necessary to consider that this food is not usually consumed as single food but it is much more frequently consumed as an ingredient in mixed pork cured meat, such as sausages, salami, mortadella, etc., which were consumed by a high proportion of cases. Unfortunately, the lack of food consumption data from healthy controls hampers drawing more specific conclusions on the magnitude of the association between PNL and PL food consumption and hepatitis E, at the individual level.

Our model was built to support risk ranking. The sensitivity analysis shows that the parameter that brings the larger uncertainty is the mean $\mu^{-1}$ of the HEV individual infectious dose distribution. This is not surprising given that the scarce availability of data to estimate μ affected the possibility to build a robust dose–response model, similar to many other exposure studies in humans. In our study, we used the total HEV load (g.e.) in food implicated in hepatitis E outbreaks as a proxy for the individual HEV infectious dose in humans. It is impossible, however, to evaluate to what degree the quantitative assessment of HEV in food differs from the true individual infectious dose. In addition, two different sources of uncertainty affect our dose–response model. On one hand, we only found four outbreak reports in the literature providing the information needed. On the other hand, the uncertainty associated with the quantitative methods used for the assessment of HEV in food should be also considered.

To obtain more reliable estimates of the actual number of HEV infections in humans, a more robust estimation of parameter μ would be needed. Nonetheless, the food ranking does not appear to be influenced by the uncertainty introduced by this specific parameter since the dose–response model acts in the same way on all types of food considered in our study. The parameters potentially introducing differences in the ranking are the ones displayed in Table 4. To have a more direct view of how these parameters affect the ranking, we have provided a qualitative risk classification by building a risk matrix. The analysis is reported in Appendix E.

Other important factors affecting the model outcomes are the mean quantity of food consumed in a year $cons_{day}$ and the mean viral concentration in food $\lambda^{-1}$. While the λ parameter was estimated from data from a large national sampling study, the $cons_{day}$ was estimated from a consumption dataset whose current reliability is difficult to assess for several reasons. First, the survey was conducted in the Italian population more than fifteen years ago and data might no longer reflect the current consumption in terms of type of food, frequency of consumption and quantity with the same level of accuracy. Second, the consumption data refer to general food categories and the lack of details on the food ingredients makes it difficult to extract the true quantities of food consumed for some categories, introducing uncertainty, in particular, for PL. Finally, estimates were only available at the national level and did not allow to take account for regional and local differences, which in the case of PL and PNL may be critical, as described above.

Another limitation of our study is the use of one single dataset for the estimation of the prevalence of HEV contamination in food. This methodological choice was driven by the lack of full comparability of the prevalence estimates among different studies due to a poor harmonization of laboratory methods used to detect HEV in food. In addition, the uncertainty analysis that we performed in our study would have been impossible using estimates from other studies. Due to the extremely high variety of foods and mode of preparation, estimating the prevalence of HEV contamination in food based on one single survey may introduce a selection bias depending on the goodness of randomization of the samples. In our study, this type of bias may be suspected for GV and SH prevalence estimation. For both these foodstuffs the estimated prevalence was 0.5% and 0%, respectively, representing a highly discrepant value compared with other similar studies conducted in Italy and abroad. In SH, Suffredini et al. and La Rosa et al. [32,33] reported much higher prevalence of HEV contamination with values ranging up to 8.1%. The prevalence of GV contamination, although considerably lower than SH, was never estimated to be 0% in other studies [18,37,38]. These considerations suggest that the role of SH and GV for HEV transmission in the Italian population may be more important than our study showed. In terms of risk ranking, however, this does not appear to substantially change our results. Different is the case of RM. This food was included in our study because it was focused as a potential risk food for HEV transmission in China in 2016, where a high prevalence of active HEV infection in cows was reported [66]. However, no further studies confirmed these findings [67,68,69].

## 5. Conclusions

Our model allowed to rank the relevance of different food categories for HEV transmission in the Italian population. In agreement with the literature and with data from Italian surveillance, pork products with and without liver emerged as the most important food implicated in HEV transmission. In particular, since our data and analyses highlight a specific risk associated with liver-containing products, actions to reduce the risk could leverage on the reduction of HEV contamination of the final products, for example, by screening livers destined to make up seasoned cured meat products or by reducing the viral load through heat treatment (e.g. pasteurization) of tissues and livers allocated to food production. Moreover, consumer information on the risk posed by liver consumption represents an important component of the mitigation strategies, in particular, in areas where the consumption of liver-containing traditional products is popular.

Another important value of our study is its contribution in shedding light into the existing data gaps that the research and surveillance activities should cover in the future to improve the risk ranking. From this perspective, the availability of a mathematical model represents a useful tool that could be further refined as new data and knowledge will be made available.

## Figures and Tables

**Figure 1 foods-11-00087-f001:**
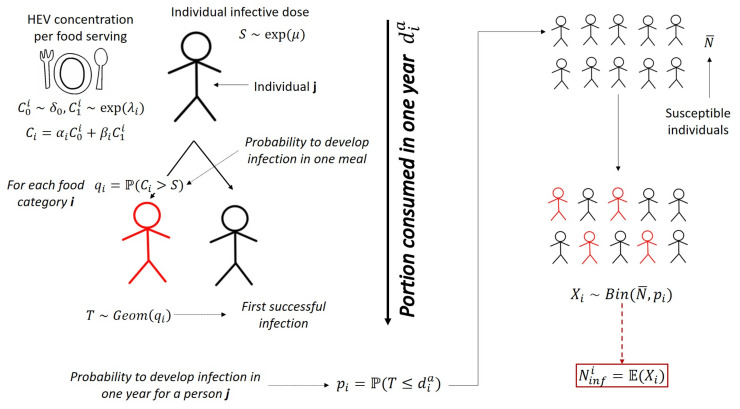
The model sketch.

**Figure 2 foods-11-00087-f002:**
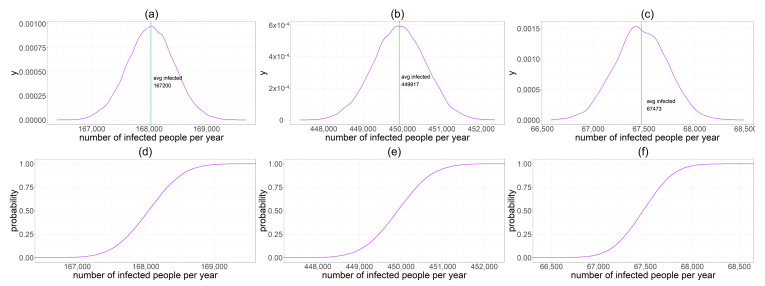
They are named from (**a**–**f**) from the top left corner to bottom right. (**a**) Density distribution of new HEV infected individuals per year $X_{PL}$. The green line indicates the mean value $N_{inf}^{PL}$. (**b**) Density distribution of new HEV infected individuals per year $X_{PNL}$. The green line indicates the mean value $N_{inf}^{PNL}$. (**c**) Density distribution of new HEV infected individuals per year $X_{SH}$. The green line indicates the mean value $N_{inf}^{SH}$. (**d**) Cumulative distribution of $X_{PL}$. (**e**) Cumulative distribution of $X_{PNL}$. (**f**) Cumulative distribution of $X_{SH}$.

**Figure 3 foods-11-00087-f003:**
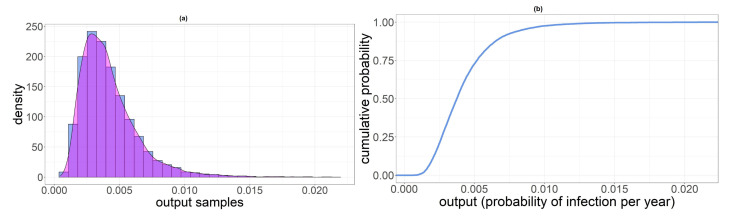
(**a**) Histogram of $p_{PL}$ samples. (**b**) Sample cumulative distribution of $p_{PL}$.

**Figure 4 foods-11-00087-f004:**
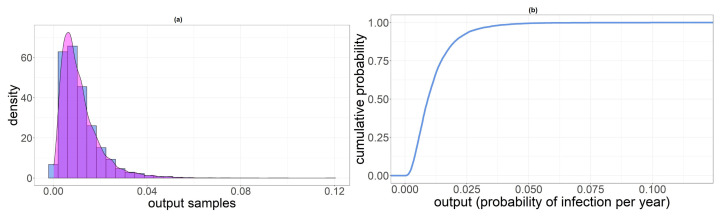
(**a**) Histogram of $p_{PNL}$ samples. (**b**) Sample cumulative distribution of $p_{PNL}$.

**Figure 5 foods-11-00087-f005:**
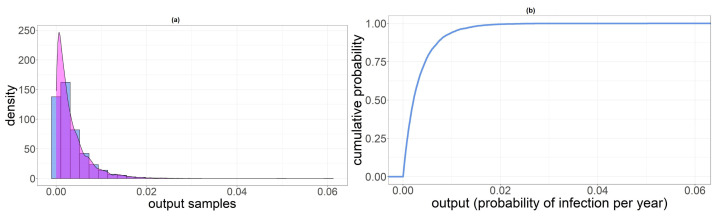
(**a**) Histogram of $p_{SH}$ samples. (**b**) Sample cumulative distribution of $p_{SH}$.

**Figure 6 foods-11-00087-f006:**
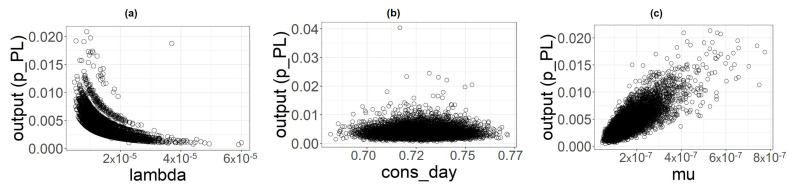
Scatterplot of input parameters for category PL versus the output considered ($p_{PL}$). (**a**) $\lambda_{PL}$. (**b**) $cons_{day}^{PL}$ (**c**) μ.

**Figure 7 foods-11-00087-f007:**
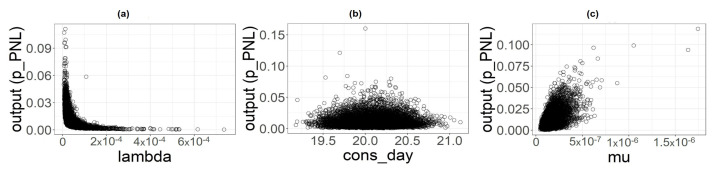
Scatterplot of input parameters for category PNL versus the output considered ($p_{PNL}$). (**a**) $\lambda_{PNL}$. (**b**) $cons_{day}^{PNL}$ (**c**) μ.

**Figure 8 foods-11-00087-f008:**
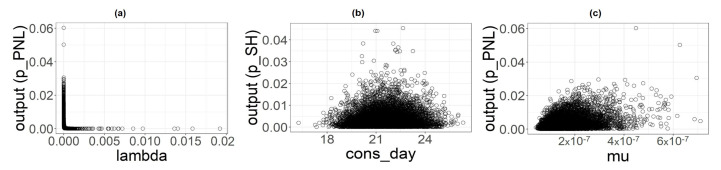
Scatterplot of input parameters for category SH versus the output considered ($p_{SH}$). (**a**) $\lambda_{SH}$. (**b**) $cons_{day}^{SH}$ (**c**) μ.

**Table 1 foods-11-00087-t001:** Model parameters.

Parameter	Description
i∈{PL, PNL, SH, GV, RM}	food category as listed in the Introduction
$\alpha_{i}=1-\beta_{i}$	contamination probability for food category *i*
$C_{i}∼\alpha_{i}\exp\left(\lambda_{i}\right)+\beta_{i}\delta_{0}$	viral concentration per serving (g.e. HEV/serving)
$\lambda_{i}^{-1}$	average viral concentration per serving (g.e. HEV/serving)
$serv_{i}$	mean serving size for food category *i* (gr)
${\left(cons_{day}^{i}\right)}^{-1}$	mean daily intake of food category *i* in a year per person (gr/day)
$d_{i}^{a}=\frac{cons_{day}^{i}\cdot 365}{serv_{i}}$	total servings consumed per year per person (number serving/year)
S∼exp(μ)	individual HEV infectious dose distribution (g.e.)
$\mu^{-1}$	mean individual infectious dose (g.e.) [13,47,48,49]
$serv_{out}$	mean serving size (gr) of food implicated in outbreaks [13,47,48,49]
*N*	Italian population 18+ (1st January 2021, ISTAT)
*h*	proportion of HEV seropositive population [50]
$\bar{N}=N\cdot \left(1-h\right)$	susceptible population
$q_{i}=P\left(C_{i}>S\right)$	probability of infection after consumption of one single serving
$T_{i}∼\mathrm{Geom}\left(q_{i}\right)$	number of failures before first successful exposure to HEV
$p_{i}=P\left(T_{i}\leq d_{i}^{a}\right)$	probability of HEV infection per individual per year
$X_{i}∼Bin\left(\bar{N},p_{i}\right)$	distribution of new HEV infected individuals per year
$N_{inf}^{i}=E\left(X_{i}\right)$	expected number of HEV infected individuals per year (no.)

**Table 2 foods-11-00087-t002:** Survey sample size.

Food Category	Sample Size
PNL	104
PL	92
RM	142
SH	204
GV	108

**Table 3 foods-11-00087-t003:** Parameter estimations per food category. A 95% CI is reported between brackets.

Category	Parameter	Estimation
PL—Liver pork products	$\alpha_{PL}$	0.11[0.06–0.20]
	$\lambda_{PL}$	$1.28\times 10^{-5}\;[6.4\times 10^{-6}$–$2.14\times 10^{-5}]$
	$serv_{PL}$	100 g
	$cons_{day}^{PL}$	0.73 $\frac{g}{day}\;[0.70$–0.752]
PNL—No liver pork products	$\alpha_{PNL}$	0.028[0.006–0.082]
	$\lambda_{PNL}$	$2.91\times 10^{-5}\;[6\times 10^{-6}$–$7\times 10^{-5}]$
	$serv_{PNL}$	100 g
	$cons_{day}^{PNL}$	20 $\frac{g}{day}\;[19.58$–20.59]
SH—Shellfish	$\alpha_{SH}$	0.0048[0.00012–0.027]
	$\lambda_{SH}$	$1.15\times 10^{-5}\;[2.91\times 10^{-7}$–$4.24\times 10^{-5}]$
	$serv_{SH}$	150 g
	$cons_{day}^{SH}$	10 $\frac{g}{day}\;[9.99$–11.7]
GV—Leafy vegetables	$\alpha_{GV}$	0[0–0.033]
	$\lambda_{GV}$	Inf
	$serv_{GV}$	100 g
	$cons_{day}^{GV}$	29 $\frac{g}{day}\;[28.62$–30.15]
RM—Raw milk	$\alpha_{RM}$	0[0–0.025]
	$\lambda_{RM}$	Inf
	$serv_{RM}$	50 g
	$cons_{day}^{RM}$	54.43 $\frac{g}{day}\;[52.77$–56.16]

**Table 4 foods-11-00087-t004:** Mean viral load per serving (g.e./gr) $E(C_{1}^{i})$, meaning $\lambda_{i}^{-1}$, the average number of serving consumed in one year $d_{i}^{a}$ and the prevalence of HEV positive food samples for each category $\alpha_{i}$.

Category	$E(C_{1}^{i})$ (g.e/serving)	$d_{i}^{a}$ (no.serving/year)	Prevalence $\alpha_{i}$
PL	78,000	3	0.11
PNL	34,000	49	0.028
SH	87,000	26	0.0048
GV	0	132	0
RM	0	107	0

**Table 5 foods-11-00087-t005:** General model parameters. A 95% CI is reported between brackets.

Parameter	Estimation
μ	$1.37\times 10^{-7}\;[5.51\times 10^{-8}$–$2.55\times 10^{-7}]$
$serv_{out}$	100 g
*h*	0.087
*N*	50,208,329
$\bar{N}$	45,840,205

**Table 6 foods-11-00087-t006:** Probability of infection following the consumption of a single serving ($q_{i}$) and in one year ($p_{i}$) and expected number of new infected individuals per year per food category in the Italian population ($N_{inf}$).

Category	$q_{i}$	$p_{i}$	$N_{\inf}^{i}$
PL	$1.22\times 10^{-3}$	$3.65\times 10^{-3}$	167,200
PNL	$1.35\times 10^{-4}$	$9.81\times 10^{-3}$	449,917
SH	$5.67\times 10^{-5}$	$1.47\times 10^{-3}$	67,473
GV	0	0	0
RM	0	0	0

**Table 7 foods-11-00087-t007:** Summary statistics of output samples.

Category	Parameter	Mean	1st Qu.	Median	3rd Qu.
PL	$p_{PL}$	$4.2\times 10^{-3}$	$2.6\times 10^{-3}$	$3.7\times 10^{-3}$	$5.1\times 10^{-3}$
	$N_{inf}^{PL}$	190,000	120,000	170,000	230,000
PNL	$p_{PNL}$	$1.1\times 10^{-2}$	$5.6\times 10^{-3}$	$9.2\times 10^{-3}$	$1.4\times 10^{-1}$
	$N_{inf}^{PL}$	520,000	250,000	420,000	670,000
SH	$p_{SH}$	$3.3\times 10^{-3}$	$8.8\times 10^{-4}$	$2.1\times 10^{-3}$	$4.5\times 10^{-3}$
	$N_{inf}^{SH}$	150,000	40,000	99,000	200,000

**Table 8 foods-11-00087-t008:** Regression analysis coefficients result for pork products containing liver.

Coefficient	Estimate	Standardized Estimate	Std. Error	*p*-Value
$\hat{\lambda_{PL}}$	$-2.2\times 10^{2}$	$-7.21\times 10^{-1}$	1.39	<$2\times 10^{-16}$
$\hat{cons_{day}^{PL}}$	$5\times 10^{-3}$	$8.13\times 10^{-1}$	$3.27\times 10^{-5}$	<$2\times 10^{-16}$
$\hat{\mu }$	$2.26\times 10^{4}$	$8.47\times 10^{-1}$	8×10	<$2\times 10^{-16}$

**Table 9 foods-11-00087-t009:** Regression analysis coefficients result for pork products without liver.

Coefficient	Estimate	Standardized Estimate	Std. Error	*p*-Value
$\hat{\lambda_{PNL}}$	−9.25×10	$-3.93\times 10^{-1}$	1.39	<$2\times 10^{-16}$
$\hat{cons_{day}^{PNL}}$	$2\times 10^{-4}$	$3\times 10^{-1}$	$7.8\times 10^{-6}$	<$2\times 10^{-16}$
$\hat{\mu }$	$7.14\times 10^{4}$	$8.76\times 10^{-1}$	$8.13\times 10^{2}$	<$2\times 10^{-16}$

**Table 10 foods-11-00087-t010:** Regression analysis coefficients result for shellfish.

Coefficient	Estimate	Standardized Estimate	Std. Error	*p*-Value
$\hat{\lambda_{SH}}$	$-1.57\times 10^{-2}$	$-1.45\times 10^{-2}$	$7.53\times 10^{-3}$	$3.6\times 10^{-2}$
$\hat{cons_{day}^{SH}}$	$1.18\times 10^{-5}$	$4.92\times 10^{-2}$	$3.96\times 10^{-6}$	$3.51\times 10^{-3}$
$\hat{\mu }$	$1.99\times 10^{4}$	$6.76\times 10^{-1}$	$4.98\times 10^{2}$	<$2\times 10^{-16}$

**Table 11 foods-11-00087-t011:** Information on food consumption among hepatitis E cases reported between 2016 and 2019 to the Italian surveillance system for acute viral hepatitis (SEIEVA).

Food	Cases with Information on Consumption Available	Cases Reporting Consumption of the Food		
	N	N	% (of Cases with Information Available)	
pork meat	90	69	76	
pork cured meat	83	58	70	
pork liver meat	51	14	29	
fruit	36	10	28	
shellfish	186	49	26	
wild boar meat	77	17	22	
vegetables	34	5	15	
wild boar cured meat	55	8	14	
offal	53	6	11	
other game meat	53	4	7	

## Data Availability

Unpublished data on viral presence in food used are not publicly available. All the other datasets used are referenced in Section 2.2. R code used for main estimation can be found here: https://github.com/Mezzanenne/Risk_ranking_Script, accessed on 1 November 2021.

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
