# Peer review of "Quantitative Methods for the Prioritization of Foods Implicated in the Transmission of Hepatititis E to Humans in Italy"

_foods, 2021, doi:10.3390/foods11010087_

Round 1

Reviewer 1 Report

Manuscript Foods-1468407 depicts a parametric stochastic model to estimate the risk of foodborne exposure to HEV. The manuscript is very well documented and presented. There is no doubt about the authors expertise to portray this mathematical model. However, based on the scope of the journal, it seems difficult to follow and corroborate the findings. The authors should provide validation process to justify the significance of the study. As presented, the findings are, to some extent, predictable and inconsequential. The manuscript could benefit considerably if the authors provide a data sheet or matrix to estimate risk using the stochastic model.

Points to be addressed,

  1. The authors should revise and correct the format to portray viral loads.
  2. A corroboration process is required.
  3. A risk estimation matrix is required.
  4. An example of food safety implementation could benefit the impact of the contribution.

Author Response

Response to Reviewer 1

[Reviewer1] Manuscript Foods-1468407 depicts a parametric stochastic model to estimate the risk of foodborne exposure to HEV. The manuscript is very well documented and presented. There is no doubt about the authors expertise to portray this mathematical model.

Thank you very much for the positive comment.

[Reviewer1] However, based on the scope of the journal, it seems difficult to follow and corroborate the findings. The authors should provide validation process to justify the significance of the study.

The comment focuses a critical point of our study. A proper validation process of the model cannot be performed due to the lack of robust external data referring to the Italian population and context. Nonetheless, we acknowledge the reviewer request on the need of some kind of corroboration of the study results. Hence, we managed to include data on food consumptions on hepatitis E cases from the Italian surveillance system for acute viral hepatitis (SEIEVA). These data have been presented and discussed, including their limitations, to corroborate the model outputs, the food ranking result and its food safety and public health significance. We would like to underline that these data were collected thanks to a a recent re-organization of the SEIEVA surveillance that was implemented starting from late 2016 to better fit the documentation of risk factors for hepatitis E.

[Reviewer1] As presented, the findings are, to some extent, predictable and inconsequential. The manuscript could benefit considerably if the authors provide a data sheet or matrix to estimate risk using the stochastic model.

We extended the model output results to include a comprehensive data sheet with the intermediate calculation of the model. In particular, we isolated the parameters directly involved in the ranking activity to help the reader individuate the main steps leading to our ranking. It should be considered that some of the point estimates reported in the table are expected number of probability distribution. The final estimations incorporate a certain amount of uncertainty, not fully explained by the point estimate reported in the table.

To address a sharper and more straightforward communication, as suggested by the reviewer, we included a risk matrix to lead the reader along the study conclusions, using the estimations and data reported in the manuscript, in particular in the aforementioned table.

We also revised importantly the discussion to extensively explain all the results and observations made connecting them to the model structure and output.

[Reviewer1] Points to be addressed,

  1. The authors should revise and correct the format to portray viral loads.

We have clarified and homogenized the notation for the viral load along the entire manuscript, hoping to address the issue raised by the referee.

  1. A corroboration process is required.

As mentioned above, we have included original data of Italian surveillance of hepatitis to corroborate our findings. Specifically, we added sections 2.2.2 and 3.5 and analysed them along the manuscript and in the discussion.

  1. A risk estimation matrix is required.

The added table with the parameters affecting the risk ranking is now included as Table4.

In Appendix E we inserted the risk matrix.

  1. An example of food safety implementation could benefit the impact of the contribution.

We acknowledge the importance of this suggestion, which practically strengthen the contribution of our study to improve food safety and public health. We have addressed this point in the conclusions with the suggestions of possible actions to reduce viral load and consumers’ food exposure to HEV.

Reviewer 2 Report

The study is interesting and has scientific merit.
I seem to be concerned about the lack of data on what criteria to select data from the literature for data extrapolation and risk modeling, especially considering that HEV is quantified by molecular methods and specific by some infectivity method. authors should deduce the limitation from the data, avoiding overestimating risks.

Author Response

Response to Reviewer 2

[Reviewer 2] The study is interesting and has scientific merit.

Thank you.

[Reviewer 2] I seem to be concerned about the lack of data on what criteria to select data from the literature for data extrapolation and risk modeling, especially considering that HEV is quantified by molecular methods and specific by some infectivity method.

Unfortunately we’re not sure to have fully caught the reviewer comment. However, we believe  the reviewer is referring to the outbreak reports that we used to extrapolate data on viral load in implicated foods. As documented in the manuscript, these are the only manuscripts allowing to directly link the viral load in the implicated food to the transmission of HEV infection to outbreak cases.   We agree on the fact that it is a delicate part of the study. We have revised the discussion to better address the assumptions and limitations connected to the use of a proxy for the estimation of the dose-response curve and the uncertainties brought by these choice. Nonetheless, all the estimates are used in a stochastic framework meaning that part of the variability and uncertainty are already explained by the probability distribution used.

[Reviewer 2] authors should deduce the limitation from the data, avoiding overestimating risks.

We acknowledge the reviewer concern about overestimating risks. However, the main purpose of our study was to generate risk ranking of food potentially implicated in the transmission of HEV to humans. In this perspective the overestimation of risks due to the limitations of data and the potential impact on the risk ranking have been further discussed, in our manuscript. In particular, we explained that uncertainty affecting the aforementioned data used to build the individual infectious dose distribution did not eventually affect the final food risk ranking, even if the number of new expected cases estimated for each single food category might be overestimated . We agree that avoiding overestimation of risks is particularly critical in case the models used for predictive purposes.

In addition we also corrected serving sizes to avoid arbitrary choices, using a 2017 guidelines of the Council for Agricultural Research and Analysis of the Agricultural Economy, authority of the Ministry of Agricultural, Food and Forestry Policies. Furthermore, we isolated the parameter that uniquely affect the ranking in the Table4 and report a qualitative risk matrix in the Appendix E. 

Reviewer 3 Report

This manuscript describes the use of a model to define the risk of specific food categories. The outcome is not surprising but does show that the selected parameters give a result akin to what we know. 

Limitations are many, but the authors do cover these well and explain in full. It is clear the model can be adapted so as more information is gathered the certainty can be improved.

Can the authors explain a little more clearly how the serving sizes were defined? Dose is a little confusing as to whether the g portion is the amount of food or if this is the expected amount of HEV? E.g., concentration of 11.6 · 104 ge/serving for PL products; so this would be the expected copy number in 150g of product? Dose i vs Dose o.

Can the number of the blood donation study participants be defined in the manuscript to relate to the population?

Author Response

Response to Reviewer 3

[Reviewer 3] This manuscript describes the use of a model to define the risk of specific food categories. The outcome is not surprising but does show that the selected parameters give a result akin to what we know. 

Limitations are many, but the authors do cover these well and explain in full. It is clear the model can be adapted so as more information is gathered the certainty can be improved.

Thank you for the encouraging comment. We revised the entire manuscript trying to address a better explanation of the study steps. In particular we included new original data on Italian surveillance of acute viral hepatitis that help to corroborate our findings. We deeply revised the methods section to give a clearer description of all the parameters, variables, and study processes. We also modified various tables in the results section to help visualize and understand the model outputs and its intermediates.  The discussion has also been fully revised to better fit the model outputs. In Appendix E we added a risk matrix to give also a qualitative acknowledgment for the risk ranking.

[Reviewer 3] Can the authors explain a little more clearly how the serving sizes were defined? Dose is a little confusing as to whether the g portion is the amount of food or if this is the expected amount of HEV? E.g., concentration of 11.6 · 104 ge/serving for PL products; so this would be the expected copy number in 150g of product? Dose i vs Dose o.

We better specified the quantities dose_i and dose_0 and explained the differences between them. We changed the names of three parameters to avoid misunderstanding and adopt clearer connotation. For this reasons  dose_i became serv_i, dose_0 became serv_out and conc_out became cons_day.

We also refined the description of the parameters in Table1.

[Reviewer 3] Can the number of the blood donation study participants be defined in the manuscript to relate to the population?

We included, as suggested, the number of participants to the blood donors study. Alongside, we reported also the size of the Italian population in the same year.
